# Dietary Aspects and Drug-Related Side Effects in Autosomal Dominant Polycystic Kidney Disease Progression

**DOI:** 10.3390/nu14214651

**Published:** 2022-11-03

**Authors:** Borja Quiroga, Roser Torra

**Affiliations:** 1Nephrology Department, Hospital Universitario de la Princesa, 28006 Madrid, Spain; 2Inherited Kidney Disorders, Department of Nephrology, Fundació Puigvert, Institut d’Investigació Biomèdica Sant Pau (IIB-SANT PAU), Universitat Autònoma de Barcelona, 08193 Barcelona, Spain

**Keywords:** autosomal dominant polycystic kidney disease, tolvaptan, aquaresis, diet, nutrition

## Abstract

Autosomal dominant polycystic kidney disease (ADPKD) is the most commonly inherited kidney disease. In the absence of targeted therapies, it invariably progresses to advanced chronic kidney disease. To date, the only approved treatment is tolvaptan, a vasopressin V2 receptor antagonist that has been demonstrated to reduce cyst growth and attenuate the decline in kidney function. However, it has various side effects, the most frequent of which is aquaresis, leading to a significant discontinuation rate. The strategies proposed to combat aquaresis include the use of thiazides or metformin and a reduction in the dietary osmotic load. Beyond the prescription of tolvaptan, which is limited to those with a rapid and progressive decline in kidney function, dietary interventions have been suggested to protect against disease progression. Moderate sodium restriction, moderate protein intake (up to 0.8 g/kg/day), avoidance of being overweight, and increased water consumption are recommended in ADPKD guidelines, though all with low-grade evidence. The aim of the present review is to critically summarize the evidence on the effect of dietary modification on ADPKD and to offer some strategies to mitigate the adverse aquaretic effects of tolvaptan.

## 1. Introduction

Autosomal dominant polycystic kidney disease (ADPKD) is the most common genetic kidney disease and the fourth most frequent renal disease leading to kidney replacement therapy in the United States [1]. Individuals with ADPKD frequently show hypertension and cardiovascular disease, independent of kidney function. Currently, there is only one approved disease-modifying drug for ADPKD, tolvaptan, and no curative treatment is available. Having experienced the disease with relatives, many patients become very frustrated at being unable to influence their disease progression.

A common approach to the management of these patients is to make them understand that the disease is inherited, but their lifestyle is in their hands and will have a notable impact on its outcome. Therefore, the goal is to try to minimize the effects of the disease and to avoid simply waiting for the genetic fate to take its course. Here, one should differentiate between healthy diets for chronic kidney disease (CKD) and those for certain specific aspects of ADPKD, in particular, and acknowledge the advertisement of unpublished diets that promise much more than they have been able to demonstrate. There are no relevant dietary intervention trials in patients with ADPKD, suggesting that patients with ADPKD should be treated differently than other patients with CKD. Miracle diets not only do not have a positive effect but can even be harmful, and worse, can even lead patients to despair. It is the task of the nephrologist to help the patient discern between dietary interventions that are medically meaningful and scientifically proven and promising diets that lack a scientific basis.

Until now, only one treatment, tolvaptan, has been approved to slow ADPKD progression. Tolvaptan is an orally active nonpeptide arginine vasopressin V2 receptor antagonist that has been demonstrated to inhibit cyst growth and, consequently, produce a modest reduction in the glomerular filtration rate slope [2,3]. However, its prescription is limited to patients with rapid and progressive decline in kidney function not only due to the lack of evidence of efficacy in other cohorts but also because of the important side effects. Indeed, the Achilles heel of tolvaptan is the significant rate of discontinuation due to its aquaretic effects (urine volume may increase to up to 8000 mL per day, with a consequent risk of dehydration) and the drug-related impacts on quality of life [2,4].

In this manuscript, we aim to review the evidence for published dietary interventions in ADPKD and the quality of life impact of tolvaptan, offering some mitigation strategies for aquaretic effects, as many of them are linked to nutritional counseling.

## 2. Effects of Dietary Modification on ADPKD Progression

### 2.1. Salt-Restricted Diet

Hypertension is the most common manifestation of ADPKD [5], and it is, therefore, not unexpected that the most prominent dietary intervention advised in patients with ADPKD is salt restriction. Furthermore, dietary sodium has been shown to influence clinical outcomes in some randomized clinical trials for other kidney diseases [6,7,8]. A post hoc analysis of the first and second Ramipril Efficacy in Nephropathy (REIN) trials suggested a potential benefit of sodium restriction [9]. In the Chronic Renal Insufficiency Cohort (CRIC) study, higher urinary sodium excretion correlated with CKD progression and all-cause mortality [10]. However, the impact of salt restriction on non-glomerular diseases such as ADPKD, although not fully proven, is supported [11,12].

The Consortium for Radiologic Imaging Studies of Polycystic Kidney Disease (CRISP) showed an association between urine sodium excretion, a surrogate marker for dietary sodium, and the rate of increase in total kidney volume at relatively early stages of the disease [13].

In the HALT-PKD trial, all participants were instructed to follow a sodium-restricted diet (≤2.4 g/day) [14,15]. A post hoc analysis suggested a causal relationship between dietary sodium and kidney growth and showed a trend toward an association between lower salt intake and a slower rate of decline in estimated glomerular filtration rate (eGFR) [12].

In the absence of trials assessing salt intake for patients with ADPKD, probably the most sensible advice to follow would be to use the previous guidelines for CKD patients [16,17] and those for the general population [18]. According to these guidelines, sodium intake in patients with ADPKD should probably be less than 2 g of sodium per day.

In summary, dietary sodium restriction allows other medications that are commonly used in ADPKD to be more effective with blood pressure reduction. Additionally, there is evidence of a detrimental effect of dietary sodium on the rate of progression of ADPKD so a sodium-restricted diet should be suggested in the management of ADPKD. As rigorous blood pressure control has been associated with better kidney outcomes [14], a multi-approach strategy should be considered to achieve lower targets (e.g., blood pressure <120/80 mmHg), combining antihypertensive drugs with a low-salt diet.

### 2.2. Protein-Restricted Diet

In ADPKD, a high protein intake can not only induce hyperfiltration but also increases vasopressin level, which might lead to cyst growth and kidney function decline [19]. Experimental data have highlighted that the early restriction of protein intake can delay CKD progression in non-ADPKD patients. Current Kidney Disease: Improving Global Outcomes (KDIGO) guidelines suggest lowering protein intake to 0.8 g/kg/day in adults with a GFR of <30 mL/min/1.73 m^2^ and avoiding intake >1.3 g/kg/day in adults with CKD at risk of progression [20]. Studies on protein restriction in patients with ADPKD, however, have failed to show benefits and have demonstrated a trend toward increased morbidity with low protein intake when eGFR falls below 24 mL/min/1.73 m^2^ [21]. On the other hand, Aukema et al. showed that both dietary protein source and dietary protein level negatively affect polycystic kidney disease progression in pcy animals [22]. The same group also demonstrated a lower increase in cystic volume in mice with low protein intake compared with mice with normal protein intake [23].

The 2015 KDIGO Controversies Conference on ADPKD did not recommend any specific protein-restricted diet for ADPKD and referred to the 2012 KDIGO guideline on CKD [24]. The updated Kidney Disease Outcomes Quality Initiative (KDOQI) guidelines for nutrition in CKD (not specific for ADPKD) recommend protein restriction even to 0.55–0.6 g/kg/day (in metabolically stable patients with CKD) [16]. Recently, Heida et al. reported a positive correlation between urine-to-plasma urea ratio and protein intake. As a high urine-to-plasma urea ratio is a sign of rapid progression, its increase by a high-protein diet appears detrimental [25]. The KHA-CARI ADPKD guidelines recommend a moderate protein diet (0.75–1.0 g/kg/day), as a low protein diet (<0.6 g/kg/day) [26].

Even in the absence of clear evidence of a positive effect of a protein-restricted diet on ADPKD, it seems wise to advise a healthy moderate protein intake of around 0.8 g/kg/day [27]. Thus, fish and meat intake should be consumed according to their protein content.

### 2.3. Hypocaloric Diet

Food restriction has been shown to decrease oxidative stress and age-related renal dysfunction and to delay the progression to later stages of CKD in the rat remnant kidney model [28,29,30]. At least two groups have shown that moderate food restriction slows disease progression in the Pkd1 and Pkd2 mouse models, potentially in a dose-dependent manner [31,32]. Warner et al. observed that even mild food restriction could significantly slow cyst progression in Pkd1 mice [28].

In addition, the benefit of fasting has been established in a reproduced non-orthologous PKD rat model (Han: SPRD), not only through time-restricted feeding (compared with ad libitum) but also with a ketogenic diet given ad libitum [31]. The effect of food restriction appears not to be mediated by changes in cyclic adenosine monophosphate (cAMP). Apparently, caloric restriction causes metabolic reprogramming. The effect of food restriction is mediated, at least in part, by a mechanism that involves the mammalian target of rapamycin (mTOR) and liver kinase B1–AMP-activated protein kinase (LKB1–AMPK) pathways [28].

Unfortunately, long-term caloric restriction is unlikely to be well-tolerated by most patients. It may also have harmful effects, such as anemia, muscle wasting, weakness, light-headedness, irritability, fatigue, and depression. A recently completed clinical trial (NCT03342742) investigated the feasibility and tolerability of a caloric deficit of 34% in 30 overweight adults with ADPKD. Time-restricted eating and intermittent fasting are being studied as alternatives to caloric restriction in patients with ADPKD (NCT 03342742, NCT 04534985). More studies in humans are needed to determine the timing and degree of caloric restriction that may provide beneficial effects in ADPKD.

Overweightness has been found to be associated with a faster rate of kidney enlargement and functional decline in early stage ADPKD [33]. Although there is insufficient evidence on this topic, it seems at least reasonable to avoid overweightness in ADPKD. On the other hand, there is not enough evidence that plant-derived protein intake may be beneficial in CKD; therefore, although restricting animal-derived protein intake may seem reasonable, there is no evidence of its benefits so far.

### 2.4. Ketogenic Diet

The reasoning behind using a ketogenic diet for ADPKD is based on the fact that cyst cells have been shown to be dependent on glucose [34], which could be prevented by a low-sugar diet such as the ketogenic one. Ketogenic diets, involving adherence to a high-fat or very low carbohydrate program, have been used for over 100 years in the management of childhood epilepsy, and ketone dietary supplements are in widespread popular use for the self-management of overweightness [35]. Ketosis can also be induced by fasting or caloric restriction. In a recent, observational study on a ketogenic diet, just over half of the patients self-enrolled in the study were able to comply with the diet for at least 6 months. Despite some objective and subjective health benefits, the study raised important questions about the long-term tolerability and safety of such interventions [36]. Tolerability failure to adhere to the ketogenic diet on a daily basis is an issue. Additionally, regarding safety, the rise in cholesterol may also put patients at a higher cardiovascular risk. Until there is strong evidence that a ketogenic diet exerts a positive effect on ADPKD outcomes, neither such a diet nor ketone supplementation should be encouraged [37].

### 2.5. High Water Intake Diet

The advice that ADPKD patients should increase water intake is based on the upregulated cAMP in ADPKD cells. Increased fluid intake prevents the action of vasopressin, and therefore, a decrease in cAMP is expected. However, the level of water intake achieved by taking tolvaptan seems unachievable by means of a high water intake diet in the absence of increased thirst. In addition to a potential effect on ADPKD progression, high water intake can help prevent nephrolithiasis, a well-known complication of the disease [38].

Small trials trying to assess the effect of increased water intake on ADPKD failed to achieve significant results [39,40]. However, there are ongoing clinical trials to demonstrate the efficacy of high water intake in patients with ADPKD (ANZCTR12614001216606, NCT02933268) [41,42]. Thus far, it has been demonstrated that the recruitment and retention of patients are feasible [40].

Even in the absence of evidence of a positive effect of high water intake, it is wise to follow KDIGO advice and increase water intake to suppress endogenous vasopressin, especially in patients not receiving tolvaptan. Although the minimum water intake has not been established, in ongoing trials, the fluid prescription is titrated in order to achieve a urine osmolality of ≤270 mOsmol/kg [41,42]. However, this recommendation should be taken with caution in the advanced stages of CKD [43].

### 2.6. Caffeine Intake

Caffeine may increase the renal concentrations of cAMP in the ADPKD cell, which is apparently an important driver of renal cyst growth. However, in small observational studies of patients with ADPKD, there was no correlation between caffeine intake and kidney outcomes [44,45]. Due to a lack of evidence, it does not seem necessary to strongly limit caffeine intake in patients with ADPKD but rather follow the World Health Organization (WHO) recommendations and limit the intake to 400 mg per day.

### 2.7. Other Nutritional Aspects

There is a lack of specific information regarding other nutritional counseling in ADPKD. Fruit and vegetables have been shown to be beneficial in patients with CKD, including early stages, and, indeed, KDOQI guidelines recommend their consumption in CKD to reduce net acid production [16,46]. In addition, bicarbonate supplementation is recommended in CKD if metabolic acidosis is present (bicarbonate target 24–26 mmol/L) [16]. Regarding mineral and bone disorders in CKD, nutritional counseling should be directed to maintain normal phosphorus levels and a neutral calcium balance [16]. However, as previously stated, there is a lack of evidence for determining the association between phosphorous levels and the development of cardiovascular and renal events [43]. Similarly, nutritional recommendations should be directed to avoid hyperpotassemia, although no specific studies have been focused on its relevance in ADPKD [16].

More controversial is the impact of dietary fiber on kidney outcomes. However, via inflammation regulation and improving microbiota, new evidence is emerging on its protective effect [47]. Likewise, the effect of alcohol consumption on these patients has not been specifically tested. However, the universal recommendations for patients with CKD and for the general population are equally valid [48].

## 3. Tolvaptan Aquaretic Effects

### 3.1. Aquaretic Effects in Clinical Trials

Pivotal clinical trials testing the attenuation of kidney function progression, namely REPRISE and TEMPO 3:4 and 4:4, demonstrated that the tolvaptan mechanism of action-derived effects was frequent and led to important rates of treatment discontinuation [2,49]. Specifically, aquaretic effects (thirst, polydipsia, polyuria, and nocturia) developed in more than 30% of the enrolled subjects. However, a recently published long-term safety study including both pivotal clinical trials showed that aquaretic effects were time-dependent [49]. Concretely, the TEMPO 4:4 trial was an open-label extension of TEMPO 3:4 including the whole cohort (initially treated patients with tolvaptan or placebo); the REPRISE trial that randomized ADPKD patients in later stages of the disease to tolvaptan or placebo has recently published a long-term extension of both study arms where patients received tolvaptan irrespective of previous treatment assignment [3]. Interestingly, in both trials, previous exposure to tolvaptan was shown to protect against adverse events (including aquaresis) in comparison to the tolvaptan-naïve cohort, with a window of susceptibility ranging from 13 to 18 months [50]. With these results, we can reassure patients about the aquaretic effects of tolvaptan while intensifying attenuation measures, especially in the first year and a half of treatment.

### 3.2. Quality of Life of Patients with Tolvaptan

In general, ADPKD is associated with poorer quality of life. Age, progression biomarkers, kidney and liver volume, and disease-specific symptoms (such as ascites or pain) have been linked to worse physical well-being [51,52,53]. However, there is scarce and heterogeneous information about the impact of the adverse effects of tolvaptan on the quality of life in ADPKD. While some specific scales have been developed to determine the quality of life in ADPKD, they are not universally used, and comparisons among the available data are subject to important bias [54,55,56,57] (Table 1).

Three small prospective studies aimed to determine the impact of aquaresis on quality of life. First, using the Kidney Disease Quality of Life Short Form (KDQOL-SF) 1.2 scale, Anderegg et al. evaluated the effect of tolvaptan on the quality of life in 98 patients with ADPKD (68 with no indication for tolvaptan and 30 with prescribed tolvaptan at stable doses) [58]. During a 1-year follow-up, 11% of the treated patients needed to discontinue tolvaptan due to aquaretic side effects. However, the tolvaptan prescription did not affect the quality of life. These intriguing results need to be understood in the context of the study design, with two-thirds of patients not receiving tolvaptan due to a medical indication and the use of strict inclusion criteria for the tolvaptan group (who were thus a select group of patients who tolerated long-term tolvaptan treatment) [59]. Tarabzuni et al. evaluated the quality of life (assessed by the Nagasaki Diabetes Insipidus Questionnaire and a questionnaire used in the evaluation of symptoms and health-related quality of life in patients with the Gitelman/Bartter syndrome) in nine patients with ADPKD who were treated with tolvaptan after a dietary intervention to reduce osmotic load. After a 3-month diet, the authors showed a negative correlation between urine volume and quality of life. In contrast, and probably due to the limitations of the study (i.e., small sample size), the dietary intervention was not associated with an improvement in quality of life [60]. Finally, psychocognitive tests were performed by Lai et al. on 36 patients with ADPKD (10 treated with tolvaptan and 26 controls), revealing lower rates of depression in the tolvaptan group [61]. Interestingly, this study also evaluated cardiovascular outcomes such as carotid intima-media thickness, epicardial adipose tissue thickness, and inflammatory parameters. Tolvaptan reduced atherosclerosis markers and inflammation, suggesting that its protective role can be partially explained by the improvement in renal function but also by the absence of overhydration due to the aquaretic effects in addition to a direct effect on extrarenal V2 receptors [61]. In terms of quality of life, a recent meta-analysis has suggested a positive effect of tolvaptan in reducing renal pain and urinary tract infections [62].

The definitive results of the ACQUIRE prospective study, which aims to explore patient-reported outcomes and health-related quality of life among patients with rapidly progressive ADPKD (NCT02848521) [57], are currently awaited. An interim analysis (over the initial 9 months), communicated during the 57th ERA-EDTA Congress, showed that disease progression negatively affects patients, especially those in the early stages of CKD [63].

### 3.3. Attenuation of Aquaretic Effects

Beyond the time-dependent attenuation of the aquaretic effects, some strategies have been proposed to reduce these effects, but the evidence of their efficacy is scarce. A recent open-label clinical trial was performed on ten patients with ADPKD who randomly received antihypertensive therapy with or without trichlormethiazide, a thiazide diuretic, for 12 weeks. A modest but significant reduction in urine volume was demonstrated with the use of trichlormethiazide, without any short-term impact on renal function (i.e., during the 12-week period) [64]. Kramers et al. recently performed a crossover trial including 13 patients with ADPKD who received sequential treatment with hydrochlorothiazide, metformin, or placebo (three 2-week periods with random order of the three treatments). Interestingly, both hydrochlorothiazide and metformin reduced urine volume after 2 weeks of follow-up (hydrochlorothiazide also improved the quality of life and biomarkers of kidney damage). In addition, a long-term follow-up of mice showed that hydrochlorothiazide reduced polyuria in tolvaptan-treated animals [65]. Although there is no strong evidence for recommending thiazides as an aquaretic effect attenuator, in a larger published observational study, Kramers et al. showed that its use was safe in terms of kidney outcomes, so it can be used at least as a second-line antihypertensive [66]. Modified-release tolvaptan was developed to achieve a once-daily prescription. The NOCTURNE phase 2 randomized, placebo-controlled trial compared the immediate-release and modified-release tolvaptan and demonstrated no differences in terms of aquaretic effects (assessed by a specific ADPKD Urinary Impact Scale), quality of life (assessed by the ADPKD Impact Scale), or patient-reported outcomes [67]. Previously, the same group designed a modest crossover trial including 25 patients with ADPKD, with similar results in terms of tolerability but with substantial interindividual variability [68].

Some dietary interventions could affect polyuria in patients with ADPKD treated with tolvaptan. In addition to blood pressure reduction, low salt intake in patients receiving tolvaptan leads to a reduction in natriuresis and consequently in diuresis volume [4]. Beyond salt reduction, from a theoretical point of view, the reduction in the osmotic load of the urine could also drive aquaretic mitigation. However, published clinical studies have not been able to demonstrate a reduction in aquaresis with a moderate protein diet (0.75–1.0 g/kg/day) [60]. Potassium intake and glucose excretion are related to the osmotic load of urine [69]. However, their effects on aquaresis have not been elucidated in tolvaptan-treated ADPKD.

Caffeine is a well-known stimulus for AMP accumulation. Until now, its effect on kidney disease progression in ADPKD has not been demonstrated. Published guidelines recommend that while excessive consumption should be avoided, consumption should be sufficient not to forego the beneficial cardiovascular effects [70]. The results regarding the effects of caffeine on natriuresis and diuresis have been controversial [71]. Stronger evidence has been acquired regarding the effect of caffeine on hypertension [72]. Overall, it would seem logical to recommend a low-sodium diet with moderate caffeine intake, especially at night, to reduce tolvaptan-related nocturia and blood pressure.

## 4. Conclusions

There are no magic diets with a great impact on the evolution of ADPKD. Despite this, nephrologists should advise a healthy diet, low in salt and without excessive protein, as well as healthy lifestyle habits, as suggested in the KDIGO guidelines for patients with CKD. While ADPKD is inherited, it is in the hands of the patient to stay in good shape and have a healthy lifestyle that will improve morbidity and quality of life.

Tolvaptan is the only approved drug for ADPKD, with evidence that it reduces disease progression. However, its aquaretic effects influence patient quality of life, leading to a significant discontinuation rate. Some strategies may be implemented to attenuate aquaresis (such as a low-sodium diet, moderate caffeine intake, or even the prescription of thiazide as an antihypertensive therapy), especially during the first 18 months, when aquaretic effects are more pronounced.

## Figures and Tables

**Table 1 nutrients-14-04651-t001:** Summary of quality of life scores in ADPKD.

Score	Items	
ADPKD-IS (period of 14 days) Scores: 1 (not difficult/bothered) to 5 (extremely difficult/bothered)	Physical domain	Engage in leisure activities Complete a full day’s work at job or home Conduct daily activities as usual, regardless of pain Complete everything in a day due to tiredness or exhaustion Perform intense physical activities Modify lifestyle due to pain or discomfort Bothered by pain
	Emotional domain	Acceptance Anxiety Sadness Feeling full before appetite satisfied
	Fatigue domain	Exhaustion or fatigue Feeling tired while driving Feeling fatigued after a good night’s sleep
ADPKD-UIS (period of 7 days) Scores: 1 (not difficult/bothered) to 5 (extremely difficult/bothered)	Urinary frequency	Daily activities Social activities Planning to use the bathroom Urination frequency
	Urinary urgency	Daily activities Social activities Planning to use the bathroom Urination urgency
	Nocturia	Ability to sleep through the night Waking up to urinate Impact on everyday life
ADPKD-PDS (period of 7 days) Scores: 1 (no pain/discomfort) to 5 (extreme pain/discomfort) Domains have an additional question: - Rate at its worst - Rate on average - Frequency	Dull kidney pain	Interfered with routine daily activities Interfered in leisure activities Interfered with relationships with other people Interfered with sleep Interfered with enjoyment of life
	Sharp kidney pain	Interfered with routine daily activities
	Fullness/discomfort	Interfered with ability to perform mild physical activities Interfered with ability to perform moderate physical activity Interfered with bending/stretching Interfered with eating or your appetite Interfered with relationships with other people

ADPKD, autosomal dominant polycystic kidney disease; ADPKD-IS, Autosomal Dominant Polycystic Kidney Disease Impact Scale; ADPKD-UIS, Autosomal Dominant Polycystic Kidney Disease Urinary Impact Scale; ADKPD-PDS, ADPKD—Pain and Discomfort Scale.

## Data Availability

Not applicable.

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
