# Peer review of "Dietary Aspects and Drug-Related Side Effects in Autosomal Dominant Polycystic Kidney Disease Progression"

_nutrients, 2022, doi:10.3390/nu14214651_

Round 1
Reviewer 1 Report
Dear sir / madam,
The authors have summarized the evidence on the effect of dietary modification on ADPKD and strategies to mitigate the side-effects of tolvaptan use. The article is well organized. However, pathways upon which these dietary interventions influence ADPKD are mostly missing and this should be addressed:
Abstract:
- Line 2: In the absence of treatment. This is incorrect, since tolvaptan is an approved drugs in ADPKD.
Introduction:
- In the introduction multiple claims are made without references. Please add references to these claims. Multiple examples below:
- The authors indicate that there is a “flourisching of diets that promise much more than they have been able to deliver” (line 38-39). Please add a reference
- The authors indicate that “the Achilles heel of tolvaptan is the significant rate of discontinuation due to its aquaretic effects and the drug-related impacts on quality of life”(line 50). Please add a reference
Salt-restricted diet
- The paragraph summarizes multiple studies in which the influence of sodium on ADPKD and non-ADPKD is investigated. However, it has not been described by which pathways sodium affects cyst growth. Please add these pathways.
- The authors indicate that “dietary sodium has been shown to influence clinical outcomes from RAAS blockade in several randomized clinical trials for other kidney diseases” (line 63). Please describe which and in what manner these clinical outcomes were influenced.
- At the end of the paragraph the authors suggest a moderate sodium restriction. However, this is not defined by the authors. Please define a moderate sodium restriction. The same argues for the use of antihypertensives. What should be prescribed as first choice.
Protein-restricted diet
- The authors mention that “dietary protein source and dietary protein level significantly affect polycystic kidney disease in pcy animals” (line 85). Please describe how polycystic kidney disease is affected.
- At the end of the paragraph the authors suggest a healthy moderate protein intake. However, this is not defined by the authors. Please define a healthy moderate protein intake.
Hypocaloric diet
- Almost at the end of the paragraph the authors write “Interestingly, Warner et al. observed that even mild food restriction could significantly slow cyst progression in Pkd1 mice” (line 113-115). This sentence seems more appropriate at the beginning this paragraph when multiple animal studies are described (line 100-105).
- Obesity/obese and overweight are being used alternately throughout the manuscript. Consider using one of these terms.
Ketogenic diet
- The authors describe a study which “raised important questions about long-term tolerability and safety of” a ketogenic diet. Please describe what important questions were raised
High water intake diet
- In the paragraph the authors describe multiple ongoing studies which investigate high water intake in ADPKD. However, no studies which have already been performed on this subject (e.g. Higashihara et. al. 2014 and El-Damanawi et. al. 2020) are described. Please consider adding these studies for the completeness of this review.
Quality of life in patients with tolvaptan
- This paragraph described the QoL in patients with tolvaptan and multiple QoL questionnaires used in ADPKD patients. It may be reasonable to describe all studies in more detail for a more adequate understanding of the clinical impact of all studies
Attentuation of aquaretic effects
- The authors describe the effect of low dietary salt intake on urinary volume in patients using tolvaptan. This effects is thought to derive from a reduction in osmotic load of the urine. Kramers et. al. also described the effects of potassium and protein intake on the osmotic load. Please consider adding some information on the effect of potassium and protein intake on urinary volume.
- At the end of the paragraph, the authors recommend a low-sodium diet. Earlier in the manuscript a moderate sodium restriction was suggested. Please recommend one or the other and define the amount of sodium which is recommended.
- Caffeine use is also mentioned in this paragraph. Is may be reasonable to add another paragraph with other interventions in ADPKD. Other dietary aspects could be mentioned, such as alcohol use and meat/fish intake.
It may be useful to add a conclusion section to this manuscript
Author Response
We would like to thank the reviewer for their constructive recommendations that we have applied. Please see our point-by-point revision below:
Abstract:
- Line 2: In the absence of treatment. This is incorrect, since tolvaptan is an approved drugs in ADPKD. RESPONSE: Thank you for the suggestion. We have change the sentence.
Introduction:
- In the introduction multiple claims are made without references. Please add references to these claims. Multiple examples below:
- The authors indicate that there is a “flourisching of diets that promise much more than they have been able to deliver” (line 38-39). Please add a reference. RESPONSE: Thanks for the observation. Unfortunately, such diets are advertised but not published. We changed the sentence accordingly.
- The authors indicate that “the Achilles heel of tolvaptan is the significant rate of discontinuation due to its aquaretic effects and the drug-related impacts on quality of life”(line 50). Please add a reference. RESPONSE: Thank you for the suggestion. Two references have been added
Salt-restricted diet
- The paragraph summarizes multiple studies in which the influence of sodium on ADPKD and non-ADPKD is investigated. However, it has not been described by which pathways sodium affects cyst growth. Please add these pathways. RESPONSE: Thank you for the commentary. To our knowledge there is no direct influence of sodium on pathways in the polycystic cell. The main damage of a high salt diet is hypertension which is damaging by itself. We only found the `provided references in the literature pointing to detrimental effect of sodium on clinical outcomes. We have added this information to the manuscript.
- The authors indicate that “dietary sodium has been shown to influence clinical outcomes from RAAS blockade in several randomized clinical trials for other kidney diseases” (line 63). Please describe which and in what manner these clinical outcomes were influenced. RESPONSE: Thank you for the commentary. Some information regarding RCT on dietary salt restriction for CKD has been provided now.
- At the end of the paragraph the authors suggest a moderate sodium restriction. However, this is not defined by the authors. Please define a moderate sodium restriction. The same argues for the use of antihypertensives. What should be prescribed as first choice. RESPONSE: We thank the reviewer asking for more precise advice. The whole paragraph has new information and a suggested intake based on CKD guidelines has been provided. However, as choice of antihypertensive drugs falls beyond the scope of this article it has not been specifically addressed.
Protein-restricted diet
- The authors mention that “dietary protein source and dietary protein level significantly affect polycystic kidney disease in pcy animals” (line 85). Please describe how polycystic kidney disease is affected. RESPONSE: Thank you for the commentary. The sentence has been slightly modified accordingly: “…negatively affect disease progression in pcy animals”
- At the end of the paragraph the authors suggest a healthy moderate protein intake. However, this is not defined by the authors. Please define a healthy moderate protein intake. RESPONSE: Thank you for the commentary. It is now defined according to WHO guidance.
Hypocaloric diet
- Almost at the end of the paragraph, the authors write, “Interestingly, Warner et al. observed that even mild food restriction could significantly slow cyst progression in Pkd1 mice” (line 113-115). This sentence seems more appropriate at the beginning this paragraph when multiple animal studies are described (line 100-105). RESPONSE: Thank you for the commentary. Thanks for pointing this out. The sentence has been moved to the beginning of the paragraph.
- Obesity/obese and overweight are being used alternately throughout the manuscript. Consider using one of these terms. RESPONSE: Thank you for the commentary. We have decided to only use overweight.
Ketogenic diet
- The authors describe a study which “raised important questions about long-term tolerability and safety of” a ketogenic diet. Please describe what important questions were raised. RESPONSE: Thank you for the commentary. Adherence and rise of cholesterol levels have now been explained as tolerability and safety issues.
High water intake diet
- In the paragraph the authors describe multiple ongoing studies which investigate high water intake in ADPKD. However, no studies which have already been performed on this subject (e.g. Higashihara et. al. 2014 and El-Damanawi al. 2020) are described. Please consider adding these studies for the completeness of this review. RESPONSE: Thank you for the commentary. The references have now been added.
Quality of life in patients with tolvaptan
- This paragraph described the QoL in patients with tolvaptan and multiple QoL questionnaires used in ADPKD patients. It may be reasonable to describe all studies in more detail for a more adequate understanding of the clinical impact of all studies. RESPONSE: Thank you for the We do not understand what the reviewer requires. The questionaries’ are detailed in table 1 and the small prospective studies are detailed immediately after the Table 1. We have added some information on the studies.
Attentuation of aquaretic effects
- The authors describe the effect of low dietary salt intake on urinary volume in patients using tolvaptan. This effects is thought to derive from a reduction in osmotic load of the urine. Kramers et. al. also described the effects of potassium and protein intake on the osmotic load. Please consider adding some information on the effect of potassium and protein intake on urinary volume. RESPONSE: Thank you for the We have added more information regarding the osmotic load as suggested.
- At the end of the paragraph, the authors recommend a low-sodium diet. Earlier in the manuscript a moderate sodium restriction was suggested. Please recommend one or the other and define the amount of sodium which is recommended. RESPONSE: Thank you for the We have unify the terminology regarding sodium recommendation.
- Caffeine use is also mentioned in this paragraph. Is may be reasonable to add another paragraph with other interventions in ADPKD. Other dietary aspects could be mentioned, such as alcohol use and meat/fish intake. RESPONSE: Thank you for the suggestion. We have added a new point 2.7 with some advises, as suggested by the reviewer.
- It may be useful to add a conclusion section to this manuscript. RESPONSE: Thank you for the suggestion. There was a conclusion section that was not uploaded in the original submission. We have added it again.
Reviewer 2 Report
This manuscript is a nice review of what exists in the literature, and therefore, provides an simplistic (and easy to understand) overview of the topic of nutrition in PKD. No new information is provided and no in depth discussion about physiologic mechanisms in play — which is not the focus/purpose of this paper. As written, it serves a review purpose.
Author Response
Thank you for the comments. Best regards.
Reviewer 3 Report
I have a significant concern about this manuscript. Dietary aspects and drug-related side effects are two different areas to be discussed. In this paper there is a very brief overview for dietary issues. For drug related side effects, quality of life was evaluated a little bit more detailly. It is kind of a summary of available trial results without any significant comparison or a key message. So, this paper has not have a structured content integrity. It may focus only on tolvaptan side effects and its impact on QoL.
The other important point is related to a previous paper published in your journal. In Nutrients, Carriazo et al published a paper on the same topic. It has significant information and Tables on various guideline statements/recommendations. However, that paper has not been mentioned about it (Carriazo S, et al. Dietary Care for ADPKD Patients: Current Status and Future Directions. Nutrients 2019). What is the reason for that?
I have also some minör concerns/suggestions.
In the abstract, the term “avoidance of excessive protein intake, “ is relatively vague, should be better reworded.
Line 64: “…..on non-proteinuric disease such as ADPKD may be questioned” non-glomerular may be more appropriate. Because even in ADPKD, there may be a certain amount of proteinuria.
Line 72-74. This statement needs to be referenced “….as rigorous blood pressure control has been associated with better kidney outcomes, a multi-approach strategy should be considered to achieve lower targets (e.g., blood pressure <120/80 mmHg), combining antihypertensive drugs with a low-salt diet.”
Regarding protein intake; KHA-CARI ADPKD guidelines recommend a moderate protein diet (0.75–1.0 g/kg/day), as a low protein diet (<0.6 g/kg/day) has not shown to slow the rate of ADPKD progression and may increase the risk of malnutrition. This should be mentioned.
Some parts of the paper is not appropriately written. One example: “Miraculous diets not only fail to exert a positive effect but also can even be harmful, and what is worse, they play on the desperation of patients.”
Author Response
We would like to thank the reviewer for their comments on our manuscript. Here are point-by-point detailed reponse to the suggestions.
- I have a significant concern about this manuscript. Dietary aspects and drug-related side effects are two different areas to be discussed. In this paper there is a very brief overview for dietary issues. For drug related side effects, quality of life was evaluated a little bit more detailly. It is kind of a summary of available trial results without any significant comparison or a key message. So, this paper has not have a structured content integrity. It may focus only on tolvaptan side effects and its impact on QoL. RESPONSE: Thank you for the suggestion. We have structured the manuscript in these two parts, as we understand that they are linked. First, dietary aspects on progression, and after, pharmacologic treatments with strategies (usually nutritional) to mitigate its adverse events. If the reviewer considers other structure, we are opened for suggestions. However, there is not enough evidence for preparing a manuscript only on QoL and tolvaptan side effects.
- The other important point is related to a previous paper published in your journal. In Nutrients, Carriazo et al published a paper on the same topic. It has significant information and Tables on various guideline statements/recommendations. However, that paper has not been mentioned about it (Carriazo S, et al. Dietary Care for ADPKD Patients: Current Status and Future Directions. Nutrients 2019). What is the reason for that? RESPONSE: Thank you for the suggestion. We have added this citation. It was not mentioned because it is a review and we prefer to use original research studies. On the other hand, our manuscript updates the evidence in dietary interventions but also is completed with QoL aspects.
- In the abstract, the term “avoidance of excessive protein intake, “ is relatively vague, should be better reworded. RESPONSE: Thank you for the suggestion. We have change the sentence.
- Line 64: “…..on non-proteinuric disease such as ADPKD may be questioned” non-glomerular may be more appropriate. Because even in ADPKD, there may be a certain amount of proteinuria. RESPONSE: Thank you for the suggestion. The wording has been changed accordingly.
- Line 72-74. This statement needs to be referenced “….as rigorous blood pressure control has been associated with better kidney outcomes, a multi-approach strategy should be considered to achieve lower targets (e.g., blood pressure <120/80 mmHg), combining antihypertensive drugs with a low-salt diet.”. RESPONSE: Thank you for the suggestion. The reference for the HALT A trial has been added in this paragraph.
- Regarding protein intake; KHA-CARI ADPKD guidelines recommend a moderate protein diet (75–1.0 g/kg/day), as a low protein diet (<0.6 g/kg/day) has not shown to slow the rate of ADPKD progression and may increase the risk of malnutrition. This should be mentioned. RESPONSE: Thank you for the suggestion. This KHA-CARI recommendation has now been added.
- Some parts of the paper is not appropriately written. One example: “Miraculous diets not only fail to exert a positive effect but also can even be harmful, and what is worse, they play on the desperation of patients.”. RESPONSE: Thank you for the suggestion. We have carefully reviewed the whole manuscript.
Round 2
Reviewer 1 Report
The authors have summarized the evidence on the effect of dietary modification on ADPKD and strategies to mitigate the side-effects of tolvaptan use. All previous recommendations have been implemented. There are still some minor points that need to be addressed:
General
- Although the level of English throughout the manuscript is good, there are some grammatical errors in the added parts. Please carefully review these parts.
Abstract:
- First line. Please change sentence to ‘most common inherited kidney disease’.
Introduction:
- Line 52: please specify these aquaretic effects. For example: due to these aquaretics effects urine production could increase up to 8000 mL per day
Salt-restricted diet
- Line 81: An abbreviations of blood pressure is used (BP) with no introduction and is not used again in de manuscript. Consider removing the abbreviation.
Protein-restricted diet
- Line 88-90: The authors mention that a high protein level could increase vasopressin levels. Please add a reference to this statement.
- Line 110: please change h/kg/day to g/kg/day
Ketogenic diet
- Line 142-144: Ketogenic diets..of overweight: please add a reference.
High water intake diet
- The authors suggest that ADPKD patients should increase their water intake. What should be an adequate water intake? A water intake that results that urine osmolality is lower than the plasma osmolality?
- ADPKD patients are more at risk for renal stone formation. Should an increase in water intake also reduce stone formation?
- At the end of this paragraph healthy eating practices are deemed reasonable. This conclusion seems out of place in this paragraph in which the focus is high water intake. Consider removing this conclusion or adding it to the final conclusion
Quality of life of patients with tolvaptan
- It is known that tolvaptan use decrease the number of acute renal pain events. In addition pain experience has a major impact on symptom burden. Please discuss whether tolvaptan use may also have a beneficial effect on quality of life.
Conclusions:
- In my opinion, diet advices in ADPKD are in line with the KDIGO guidelines for patients with CKD, except for the increase in water intake.
- This should be mentioned more explicit. In addition, please specify in the conclusion section some strategies (line 295).
Author Response
We want to thank the reviewer for her/his time on revising our manuscript that have clearly improved. Here is detailed a point-by-point response for each recomendation.
The authors have summarized the evidence on the effect of dietary modification on ADPKD and strategies to mitigate the side-effects of tolvaptan use. All previous recommendations have been implemented. There are still some minor points that need to be addressed:
General
- Although the level of English throughout the manuscript is good, there are some grammatical errors in the added parts. Please carefully review these parts. RESPONSE: Thank you for the suggestion. A native English speaker has revised the manuscript.
Abstract:
- First line. Please change sentence to ‘most common inherited kidney disease’. RESPONSE: Thank you for the suggestion. We have modified the sentence.
Introduction:
- Line 52: please specify these aquaretic effects. For example: due to these aquaretics effects urine production could increase up to 8000 mL per day. RESPONSE: Thank you for the suggestion. We have modified the sentence.
Salt-restricted diet
- Line 81: An abbreviations of blood pressure is used (BP) with no introduction and is not used again in de manuscript. Consider removing the abbreviation. Thank you for the suggestion. RESPONSE: Thank you for the suggestion. We have removed the abbreviation.
Protein-restricted diet
- Line 88-90: The authors mention that a high protein level could increase vasopressin levels. Please add a reference to this statement. RESPONSE: Thank you for the suggestion. We have added a reference for the statement.
- Line 110: please change h/kg/day to g/kg/day. RESPONSE: Thank you for the suggestion. We have corrected the typo.
Ketogenic diet
- Line 142-144: Ketogenic diets..of overweight: please add a reference. Thank you for the suggestion. We have added a reference for the statement.
High water intake diet
- The authors suggest that ADPKD patients should increase their water intake. What should be an adequate water intake? A water intake that results that urine osmolality is lower than the plasma osmolality? RESPONSE: Thank you for the suggestion. We have added a sentence with the recommendation based on DRINK and PREVENT-ADPKD trials.
- ADPKD patients are more at risk for renal stone formation. Should an increase in water intake also reduce stone formation? RESPONSE: Thank you for the suggestion. This is an interesting point to hypothesize. We have added information on this regard in the manuscript.
- At the end of this paragraph healthy eating practices are deemed reasonable. This conclusion seems out of place in this paragraph in which the focus is high water intake. Consider removing this conclusion or adding it to the final conclusion. RESPONSE: Thank you for the suggestion. We have remove this sentence. It is a sentence in the conclusion section suggesting a healthy diet.
Quality of life of patients with tolvaptan
- It is known that tolvaptan use decrease the number of acute renal pain events. In addition pain experience has a major impact on symptom burden. Please discuss whether tolvaptan use may also have a beneficial effect on quality of life. RESPONSE: Thank you for the suggestion. We have added information based on a recent meta-analysis with the effect of tolvaptan in pain and also in urinary track infections.
Conclusions:
- In my opinion, diet advices in ADPKD are in line with the KDIGO guidelines for patients with CKD, except for the increase in water intake. This should be mentioned more explicit. In addition, please specify in the conclusion section some strategies (line 295). Thank you for the suggestion. We have completed the conclusion section with the information suggested by the reviewer.
Best regards and thank you again,
Borja Quiroga.
Reviewer 3 Report
Corrections were found reasonable.
Author Response
Thank you for the positive feedback.
Best regards, Borja Quiroga.